# Improving Obesogenic Dietary Behaviors among Adolescents: A Systematic Review of Randomized Controlled Trials

**DOI:** 10.3390/nu14214592

**Published:** 2022-11-01

**Authors:** Elodie Nonguierma, Emily Lesco, Regan Olak, Hunter Welch, Nagina Zar Alam, Jamila Bonyadi, Laura Hopkins

**Affiliations:** 1MetroHealth System, Cleveland, OH 44109, USA; 2Department of Public Health and Prevention Science, College of Education and Health Sciences, Baldwin Wallace University, Berea, OH 44017, USA; 3Department of Health Sciences and Human Performance, Cleveland State University, Cleveland, OH 44115, USA

**Keywords:** obesogenic, dietary behaviors, dietary intake, adolescent

## Abstract

The overweight and obesity epidemic persists, and over 340 million children and adolescents aged 5–19 were classified as overweight or obese worldwide in 2020. Obesity intervention becomes crucial during the adolescent years due to the increased autonomy and adolescent motivation to oversee one’s own behaviors and lifestyle-related decisions. The objective of the current study was to conduct a systematic review of randomized controlled trials aimed at improving dietary intake and behaviors among adolescents. The Preferred Reporting Items for Systematic Reviews and Meta-Analysis (PRISMA) guidelines were utilized. The key terms used in the searches referred to the study population and the topic of interest and included words and phrases such as “obesity or overweight and adolescents”, “dietary behaviors and adolescents”, “dietary intake and adolescents”, and “dietary habits and adolescents.” A tertiary screening process was employed, and the National Heart, Lung, and Blood Institute Quality Assessment of Controlled Intervention Studies quality scoring tool was utilized to assess the quality of research articles independently by *n* = 2 researchers. A total of 7441 articles were identified through the database search, and 36 were included in the current systematic review. The most common outcomes explored included dietary behaviors, anthropometric or biometric outcomes, and physical activity. Approximately half of the studies demonstrated significant improvements in the primary outcomes investigated. The majority of the high-impact studies where significant improvements in primary outcomes were demonstrated were conducted in school settings or were multicomponent or multilevel in nature. Thus, interventions targeting dietary behaviors in adolescents that are delivered in the school setting and are multicomponent or multilevel in nature are the most effective in terms of impact on dietary intake, anthropometric or biometric outcomes, and physical activity.

## 1. Introduction

Overweight and obesity continue to plague the globe, with rates having tripled since 1975 [1]. In 2020, the World Health Organization reported that 39 million children under the age of 5 and 340 million children and adolescents aged 5–19 were classified as overweight or obese [1]. While developed countries have the highest prevalence rates of childhood obesity, it is important to acknowledge obesity trends in developing countries, as the pervasiveness of childhood and adult obesity alike is increasing, creating a dual burden of malnutrition. While overweight and obesity are often primarily driven by obesogenic behaviors such as dietary overconsumption and physical inactivity, these behaviors are typically a result of socioeconomic and environmental circumstances that create unfavorable conditions to support healthy behaviors.

Adolescent obesity is associated with numerous consequences that can affect all aspects of an individual’s health. Studies suggest that adolescent obesity is associated with a myriad of negative health outcomes, including reductions in the health-related quality of life, poor self-concept, and behavioral difficulties [2]. The long-term effects of adolescent obesity include significant increases in the risk of cardiovascular diseases, cancer, and diabetes, as well as enhanced difficulty in altering poor dietary behaviors [3]. However, the emotional impacts of childhood obesity are less commonly discussed. In a systematic review of studies examining the emotional and psychological consequences of being overweight or obese as a child, researchers found that younger children, girls, and adolescents encounter significant consequences as a result of being overweight. Furthermore, some findings suggest a possible relationship between obesity and behavioral problems in children observed by parents and teachers [4]. 

Obesity intervention becomes crucial during the adolescent years due to the increased autonomy and adolescent motivation to oversee one’s own behaviors and lifestyle-related decisions [2]. While adolescents are gaining this newfound autonomy, they are simultaneously entering a developmental period associated with heightened perceived stress. This developmental shift can manifest in various unhealthy eating habits, such as an increased reliance on food as a reward and stress eating, which can lead to excessive weight gain [5]. Intervening during this sensitive time period could aid in preventing excess weight gain and therefore encourage better health outcomes for adolescents throughout their development. Data demonstrate that youth who are obese during their teenage years have an over 90% likelihood of being overweight or obese until 35 years of age, so intervention in this time period is critical for future health and quality of life [6].

Given the persisting overweight and obesity epidemic that plagues the globe, and since adolescence is a developmental period that presents unique opportunities to intervene and influence personal, independent behaviors such as dietary intake and other behaviors that impact weight status (e.g., physical activity and sleep), it is imperative that the scientific community understands the strategies that can positively impact weight status among adolescents. While implementation at the community, institutional, and policy levels can be challenging, randomized controlled trials (RCTs) are still the hallmark of medical and public health research in terms of determining the effectiveness of interventions. Therefore, the objective of the current study was to conduct a systematic review of RCTs that aimed to improve dietary intake and behaviors among adolescents. 

## 2. Materials and Methods

### 2.1. Search Strategy and Study Selection

The Preferred Reporting Items for Systematic Reviews and Meta-Analysis (PRISMA) guidelines were utilized to conduct the current systematic review [7]. The databases used included PubMed/Medline, CINAHL, PsycINFO, Cochrane Library, ClinicalTrials.gov, and Web of Science. The key terms used in the searches referred to the study population and topic of interest and included words and phrases such as “obesity or overweight and adolescents”, “dietary behaviors and adolescents”, “dietary intake and adolescents”, and “dietary habits and adolescents”. 

### 2.2. Inclusion and Exclusion Criteria

The database search was limited to studies available in English and involving human subjects. To be included, studies had to be RCTs published in the last decade (from 1 April 2012 to 1 April 2022). Other inclusion criteria were: (1) study population included participants in the adolescent age range (defined as 10 to 19 years of age; and (2) intervention activities were related to dietary behaviors, dietary intake, and dietary habits. Studies were excluded if they targeted a population with a specific health condition that may impact dietary intake, e.g., those with disordered eating or cancer. Additionally, studies were excluded if their sole focus was on physical activity behaviors without any intervention activities related to dietary or nutritional intake. 

### 2.3. Data Extraction

The National Heart, Lung, and Blood Institute (NHLBI) Quality Assessment of Controlled Intervention Studies quality scoring tool was utilized to assess the quality of research articles [8]. Two independent researchers assessed the quality of each article. The Controlled Intervention Studies scoring tool is a 14-item, yes-or-no instrument. Through the provision of affirmative (Yes) or negative (No) responses to each item, scorers rate the quality of the article as ‘poor’, ‘fair’, or ‘good.’ In situations where scoring discrepancies occurred, the senior researcher (LH) discussed the article’s score with the two independent researchers and came to a consensus regarding inclusion or exclusion. Articles that received a rating of ‘good’ were included in the current systematic review. Data were extracted and organized in tabular format (see Table 1). 

## 3. Results

### 3.1. Study Inclusion

A total of 7441 articles were identified through the database search. After the removal of duplicates, 6940 articles were screened. A tertiary screening approach was utilized, whereby abstracts were first screened based on inclusion and exclusion criteria via a review of titles (*n* = 6584 excluded), then a review of abstracts (*n* = 224 excluded). The remaining *n* = 132 articles were read entirely to review whether they met the inclusion and exclusion criteria (*n* = 40 excluded). A total of *n* = 74 articles were rated on quality using the NHLBI Quality Assessment of Controlled Intervention Studies quality scoring tool (described above). Thirty-eight articles were excluded for a poor or fair quality rating, resulting in a total of *n* = 36 research articles included in the current systematic review. In four instances, two research articles were longer-term follow-up results of another included research article. Therefore, a total of 31 independent research studies were examined in the current systematic review. See Figure 1 for the PRISMA diagram.

### 3.2. Study Design and Intervention Characteristics 

Table 1 shows the characteristics and relevant findings of the studies included in the current analysis. Most of the studies were conducted in North America, [5,9,10,11,12,13,14,15,16,17,18,19,20,21,22,23,24], with the United States as the most frequent country. Few studies were conducted in Asian [25,26,27,28,29,30], Oceanic [31,32,33,34,35], South American [36,37,38], or European [39,40,41,42,43] countries. The main intervention settings were school [12,13,14,17,18,20,25,26,27,28,29,31,32,34,35,37,38,39,41,42,43], clinical [5,10,11,24,30,40], home [9,21,22,23,33,36], and online [15,19]. In addition, a few studies were reported as multi-setting programs [9,13,14,36]. 

All the studies included in the systematic review were RCTs, and the number of participants from each study varied widely from 31 [42] to 3110 [39]. The focus age range was also wide, including children from as early as 5 [33] to 20 [42] years old, but all studies included the adolescent age range. Of the 36 included studies, two main intervention approaches were identified: nutrition education interventions and multicomponent interventions. The nutrition education intervention approach was defined as the delivery of health programs that included education on nutrition or dieting [5,11,12,15,16,17,18,21,22,23,25,26,27,28,29,30,37,38,41]. Multicomponent interventions included not only a nutrition education component, but also an additional component such as physical activity [9,10,13,14,19,20,24,31,32,33,34,35,36,39,40,43] or overall wellbeing, stress reduction, and social relationships [10,42]. Over half of the interventions were also multilevel in nature, meaning that multiple levels of the socioecological model, e.g., families or peers, school environments, and home environments, were engaged or manipulated through intervention activities [9,13,14,17,18,21,22,23,24,25,27,28,29,31,34,35,36,37,38,42,43]. The duration of the included studies ranged from 5 weeks [17,18] to 28 months [38]. Exposure time was not reported in most studies and therefore could not be compared across studies. 

### 3.3. Effect of Interventions

#### 3.3.1. Dietary Education Interventions

Nineteen of the included articles investigated the effect of nutrition education interventions on healthy eating, as well as other related outcomes, among adolescents [5,11,12,15,16,17,18,21,22,23,25,26,27,28,29,30,37,38,41]. These interventions included weightgainprevention education, nutritional education, and counseling. Eight studies reported significant differences between the control and intervention groups in terms of fruit consumption, vegetable consumption, diet quality, snacking habits, food reward, and stress eating [5,18,22,23,26,27,38,41]. Some of the included studies explored changes in the body mass index (BMI) and waist circumference and the prevention of excess weight gain, and significant improvements among intervention participants were observed [12,17,27,29,38]. Factors such as knowledge, attitude, self-efficacy, social support, intention, situation, meal timing, healthy diet/exercise behavior points, and parental behavioral strategies were also considered [22,23,25,28,29]. 

As stated above, eight studies demonstrated significant impacts on dietary behaviors, specifically fruit and vegetable intake, diet quality, and snacking habits. Bogart et al. [18] implemented a school-based obesity-prevention intervention combining school-wide environmental changes, multimedia encouragement to eat healthy school cafeteria foods, and peer-led education. As a result, the intervention group showed significant increases in the proportion of students consuming fruit and lunch and a significant decrease in the proportion of students buying snacks at school. Bessems et al. [41] implemented an eight-lesson educational intervention and observed significantly improved short- and long-term changes in fruit consumption among the intervention participants compared to the control participants. While no intervention effect was observed for BMI, Cunha et al. [37] observed significant improvements in fruit consumption, as well as significant reductions in sugar-sweetened beverage and cookie consumption. Hidayanty et al. [27] observed decreased unfavorable snacking habits in the experimental group of overweight adolescents. Horton et al. [22] and Arredondo et al. [23] explored the impact of a ‘promatora’-delivered family-meals intervention in the home. Intervention group participants demonstrated a reduced weekly consumption of fast food, as well as a greater variety of consumed vegetables at 10 months post-intervention. The study of Keshani et al. [26] aimed to assess the impact of educational intervention, based on the health belief model (HBM) and collaborative learning techniques, on diet quality in adolescents, and diet quality improved in the experimental group compared to the control group. Shomaker et al. [5] observed lower food reward and a reduction in stress eating among intervention participants compared to control participants. The study by Ochoa-Avilés [38] aimed at improving the dietary intake and physical activity of Ecuadorian adolescents. As a result, participants from the intervention group consumed lower quantities of unhealthy snacks and less added sugar at the end of the trial. Daily fruit and vegetable intake decreased in both the intervention and control groups compared to baseline. 

Several studies have demonstrated the effects of interventions through the observation of significant improvements in anthropometric outcomes. While significant between-group differences were not observed, Bagherniya et al. [29] found significant mean BMI and waist circumference reductions in the intervention group. Bogart et al. [17,18] demonstrated significant reductions in BMI over time among all participants, with significant intervention effects observed for students who were obese at baseline. Ochoa-Avilés et al. [38] observed a reduction in waist circumference in the intervention group at the end of the program. The study by Daly et al. [12] examined the effects of a satiety-focused mindful eating intervention (MEI) on BMI, weight, and mindful awareness, and observed a decrease in BMI in MEI participants compared to the control group participants. Hidayanty et al. [27] aimed to employ social cognitive theory to reduce snacking habits and sedentary activity among overweight adolescents and observed a higher reduction in BMI z-scores and waist circumference among intervention participants.

In their HBM intervention, Keshani et al. [26] observed improvements in not only diet quality but also all HBM factors and knowledge in the experimental group compared to the control group. Bagherniya et al. [29] observed that intervention participants had significantly improved self-efficacy, social support, intention, and situation compared to control participants. In their ‘promatora’ intervention, Arrendando et al. [23] observed that parent-reported behavioral strategies mediated the effects of the intervention on child dietary outcomes. 

#### 3.3.2. Multicomponent Interventions

Seventeen of the identified studies used a multicomponent intervention method to facilitate their programs [9,10,13,14,19,20,24,31,32,33,34,35,36,39,40,42,43]. Fourteen of the seventeen studies demonstrated significant improvements in anthropometric or biometric measures [10,24,34,36,39,43], dietary outcomes [13,14,19,20,32,34,35,39], or physical activity outcomes [13,14,19,20,32,39].

Significant improvements in the respective measures of BMI were observed by Anderson et al. [33], Barnes et al. [34], DeBar et al. [24], Friera et al. [43], Vidmar et al. [10], and Viggiano et al. [39]. Additional anthropometric and biometric measures were explored across the studies. Barnes et al. [34], Friera et al. [43], and Viggiano et al. [39] additionally observed significant improvements in waist circumference. Friera et al. [43] additionally observed significant decreases in percent fat mass and systolic and diastolic blood pressure and increases in percent muscle mass at both 3 and 6 months. Contrary to the expected outcomes, Sgambto et al. [36] observed significant increases in BMI among the intervention participants compared to the control participants. Additionally, participants who received both in-school and at-home intervention components showed significantly increased percent body fat compared to the control participants. However, male intervention participants demonstrated a significantly greater decrease in percent body fat.

A multitude of dietary outcomes were assessed across the included research studies, and significant findings were observed in eight of the studies [13,14,19,20,32,34,35,39]. Collins et al. [32] observed greater water intake and a greater proportion of female intervention participants consuming less than one sugar-sweetened beverage at 12 months. Cullen et al. [19] observed a significant increase in vegetable intake among intervention participants. Pberts et al. [13,14] observed significant dietary outcomes at 2, 6, and 8 months. At 2 months, compared to control participants, intervention participants ate breakfast on significantly more days per week, had a lower intake of total sugar, and had a lower intake of added sugar [13]. At 6 months, compared to control participants, intervention participants were more likely to drink soda less than or equal to once per day and eat at a fast-food restaurant less than or equal to once per week [13]. Finally, at 8 months, intervention participants reported eating breakfast on more days per week [14]. Sutherland et al. observed that intervention participants’ mean lunchbox energy from recommended foods increased significantly. Viggiano et al. [39] observed significant dietary improvements with regard to the Adolescent Food Habit Checklist at 6 months and 18 months, as well as improved nutrition knowledge and healthy and unhealthy diet food habits as recorded by the dietary questionnaire utilized. Whittemore et al. [20] observed significant improvements in self-efficacy, healthy eating behavior, fruit and vegetable intake, sugar-sweetened drink intake, and junk-food intake among participants.

Similar to dietary outcomes, a variety of outcomes related to physical activity were measured across the studies. Significant improvements were observed for screen time, resistance-training skill competency, self-reported physical activity, and moderate and vigorous exercise. However, Cullen et al. [19] noted that the control participants reported greater physical activity enjoyment.

**Table 1 nutrients-14-04592-t001:** Characteristics of included studies.

Study: First Author and Year	Country and Setting	Participant Age and Sample Size	Intervention Description	Outcomes Measured	Relevant Results
Nutrition Education Interventions					
Bagherniya 2022 [29]	Iran School	7th and 8th Grade Enrolled (*n* = 172) Intervention (*n* = 87) Control (*n* = 85)	Intervention—7 months rooted in social cognitive theory, including students (2× per month) and parents (1× per month): (i) Practical nutrition workshops; (ii) Interactive seminar. Control group: Both the students and their parents received six lectures and two nutritional books.	Body mass index (BMI) Waist circumferance (WC) Psychological variables (self-efficacy, social support, intention, and situation)	No significant between-group differences for changes in BMI (*p* = 0.13) and WC (*p* = 0.40) at 7 months. Mean BMI and WC reduced (*p* < 0.001) in the intervention group. Compared to the control group, intervention participants significantly improved their dietary behaviors and most psychological variables (self-efficacy, social support, intention, and situation) (*p* < 0.05).
Bessems 2012 [41]	Netherlands School	12–14 years old Intervention (*n* = 1117) Control (*n* = 758)	Intervention—*Krachtvoer*: (i) Eight educational lessons focusing on the consumption of fruit, a daily healthy breakfast, and decreasing the consumption of fats with low-fat snack replacements; (ii) Optional activities.	Average fruit consumption per day Weekly breakfast consumption Weekly snack consumption between meals and number of times per day	Intervention participants showed significant positive short- and long-term changes in fruit consumption (*p* = 0.048 and *p* = 0.033) compared to control participants.
Bogart 2014 [18]	USA School	12–13 years old Baseline Data (*n* = 2439) Intervention (*n* = 1178) Control (*n*-1261)	5-week intervention, including: (i) School-wide environmental changes; (ii) Multimedia encouragement to eat healthier cafeteria food; (iii) Student advocacy.	BMI Cafeteria attitudes Water attitudes Knowledge about healthy eating and physical activity Water consumption frequency	Compared to control schools, intervention schools served more fruit (*p* = 0.006), more lunches (*p* < 0.001), and fewer snacks (*p* < 0.001). Compared to control-school students, intervention-school students reported more positive attitudes towards cafeteria food (*p* = 0.02) and tap water (*p* = 0.03), greater obesity-prevention knowledge (*p* = 0.006), increased intentions to drink water from the tap (*p* = 0.04) or a refillable bottle (*p* = 0.02), and greater tap-water consumption (*p* = 0.04).
Bogart 2016 [17] Two-year follow-up of Bogart 2014		Baseline survey and anthropometric data and 2-year follow-up anthropometric data (*n* = 1368) Intervention (*n* = 829) Control (*n* = 539)			Students overall, overweight students, and obese students across groups demonstrated significant decreases in BMI overtime; no significant intervention effect was observed for intervention vs. control students overall. Among students who were obese, a significant intervention effect for improved BMI was observed (*p* = 0.005).
Cunha 2013 [37]	Brazil School	Mean age: 11 years Enrolled (*n* = 559) Intervention: 10 classes with *n* = 277 Control: 10 classes with *n* = 282	Intervention: (i) 9 nutritional education sessions; (ii) Parents and teachers received reinforcing information.	BMI Percent body fat	Intervention effects on BMI were not observed (*p* = 0.75). Intervention participants demonstrated significant reductions in sugar-sweetened beverage (*p* = 0.02) and cookie (*p* < 0.001) consumption and a significant increase in fruit consumption (*p* = 0.04).
Daly 2016 [12]	USA School	14–16 years old Enrolled (*n* = 37)	Mindful eating intervention (MEI): six weekly, 90 min curricular sessions. Control: written diet and exercise information.	BMI Mindful awareness was measured using the Mindful Attention Awareness Scale Motivation for participation questionnaire	MEI participants showed significantly decreased BMI compared with control group (CG) participants, (*p* < 0.001).
Demir 2019 [28]	Turkey School	Sixth and seventh grades Enrolled (*n* = 76) Intervention (*n* = 38) Control (*n* = 38)	Intervention: (i) 5 creative drama sessions with teen groups; (ii) Mothers received one session of training.	BMI Body weight Body height Waist circumference	Intervention participants significantly improved knowledge, attitude, meal timing, and healthy diet/exercise behavior points (*ps* < 0.05). Intervention participants demonstrated significant decreases in BMI, body weight, and waist circumference measurements (*ps* < 0.05). Control participants demonstrated significant increases in waist circumference and body weight measurements (*ps* < 0.05).
Ebbeling 2012 [21]	USA Home	14–16 years old Enrolled (*n* = 224) Intervention (*n* = 105)	1-year intervention consisted of: (i) Home delivery of noncaloric beverages (e.g., bottled water and “diet” beverages) every 2 weeks; (ii) Monthly motivational telephone calls with parents (30 min per call); (iii) Three check-in visits with participants (20 min per visit).	BMI	At 1 year, there were significant between-group differences for changes in BMI (*p* = 0.045). Change in mean BMI at 2 years did not differ significantly between the two groups (*p* = 0.46).
Ghasab 2019 [25]	Iran School	13–15 years Enrolled (*n* = 230) Intervention group (*n* = 115) Control group (*n* = 115)	The multicomponent intervention comprised: (i) Developing separate nutrition education packages for adolescents, parents, and teachers; (ii) Forming a supportive group consisting of parents and teachers to encourage the adolescents’ healthy nutritional behaviors; (iii) Holding meetings with decision-makers and stakeholders; (iv) Providing participatory homework involving parents and adolescents; (v) Sending nutritional messages to the parents.	A 60-item questionnaire was developed by the research team including 3 sections: (i) Demographic characteristics (age and parents’ age, occupation, education level, and monthly income); (ii) Dietary behavior determinants (Australian adolescent healthy eating behaviors questionnaire). Dietary behaviors (the World Health Organization Global School-based Student Health Survey questionnaire and the Youth Risk Behavior Survey)	At 3 and 6 months, intervention participants demonstrated significantly greater increases in all behavioral determinants compared to control participants (*ps* < 0.001). Breakfast, fruit, vegetable, snack, and fast-food consumption improved significantly among intervention participants compared to control participants (*ps* < 0.001).
Hidayanty 2016 [27]	South Sulawesi School	11–15 years old Enrolled (*n* = 238) Intervention (*n* = 118) Control (*n* = 120)	*Healthy Life Program* (HLP) intervention: (i) 12 weekly 75 min nutrition education sessions; (ii) Parents received nutrition education leaflets weekly for 12 weeks. Control participants received handouts of evidenced-based nutrition information based on guidelines in Indonesia.	BMI Waist circumference Snacking habits (Food Frequency Questionnaire (FFQ)) Sedentary activity (Adolescent Sedentary Activity Questionnaire) Self-efficacy in decreasing snacking and sedentary activity questionnaire	At 3 months, compared to control participants, intervention participants demonstrated a higher reduction in BMI z-scores (*p* < 0.05) and waist circumference (*p* < 0.05), decreased snacking habits (*p* < 0.05), and improved self-efficacy (*p* < 0.05).
Horton 2013 [22]	USA (US–Mexico border, Imperial County, CA) Home	7–13 years old Enrolled (*n* = 361) Intervention (*n* = 181) Control (*n* = 180)	*Entre Familia* intervention: (i) 4-month *promotora*-delivered sessions; (ii) Child-focused activities; (iii) Weekly family tasks.	Child intake (child-reported) Daily fruit and vegetable intake (National Cancer Institute Food Attitudes and Behavior survey) Monthly varieties of fruits and vegetables Daily servings of sugar-sweetened beverages (Youth/Adolescent Questionnaire) Weekly fast-food consumption Parent intake (parent-reported daily servings of fruits and vegetables) (National Cancer Institute Fruit and Vegetable All-Day Screener) Percent energy from fat (National Cancer Institute’s Multifactor Fat Screener) Parenting strategies (Parenting Strategies for Eating and Activity Scale (PEAS)) Parent-reported dietary behavioral strategies: (i) Behavioral strategies to increase fiber intake (4-point Likert scale); (ii) Behavioral strategies to decrease dietary fat intake (4-point Likert scale).	Intervention group showed reduced weekly consumption of fast food (*p* < 0.05). At 10 months, compared to control group, intervention group participants demonstrated significant differences in monthly varieties of vegetables (*p* = 0.03). Parent-reported behavioral strategies to increase fiber and lower fat intake mediated the relationship between the intervention and children’s intake of varieties of vegetables (*p* = 0.05). Parents’ percent energy from fat and behavioral strategies to lower fat intake were mediators of children’s daily servings of sugar-sweetened beverages (*p* = 0.05).
Arredondo 2019 [23] 10-month post-baseline from Horton et al. (2013)					
Keshani 2019 [26]	Iran School	13–15 years old Enrolled (*n* = 311) Intervention (*n* = 163) Control (*n* = 148)	Intervention: (i) Four 90-min interactive learning sessions; (ii) Briefing session for parents and school instructors.	Dietary intake (168-item FFQ) Revised children’s diet quality index (RCDQI) Health belief model (HBM) factors—knowledge, perceived benefits and barriers, self-efficacy, perceived severity, perceived susceptibility, and cues to action	Compared to control participants, intervention participant scores for RBQI (*p* < 0.001) and component scores of sugar (*p* < 0.001); fat (linoleic acid (*p* < 0.001) and linolenic acid (*p* < 0.001)); dairy (*p* < 0.001); fruits (*p* < 0.001); vegetables *p* < 0.001); and energy intake (*p* < 0.001) significantly improved. Compared to control participants, intervention participant HBM factors significantly improved: knowledge (*p* < 0.001), perceived benefits and barriers (*p* < 0.001), self-efficacy (*p* = 0.001), perceived severity (*p* < 0.001), perceived susceptibility (*p* < 0.001), and cues to action (*p* = 0.002).
Lee 2020 [30]	Korea Clinical	6–17 years old Mean age: 10.95 years Enrolled (*n* = 104) Intervention (*n* = 54) Control (*n* = 50)	Control (usual care group (UG)) (both UG and NG): 6 nutrition education sessions. Intervention (nutrition group (NG)). Individualized nutrition care process (diagnose, intervene, monitor, and evaluate) approach.	Changes in diet quality (Diet Quality Index International (DQI-I) and high-calorie, low-nutrient (HCLN)) Changes in BMI z-score (age and sex standardized body mass index) Intake of macronutrients (energy, carbohydrates, fat, and protein) Self-efficacy	DQI-I score also increased with respect to sodium (*p* < 0.001). NG self-efficacy increased (*p* < 0.01). At 24 weeks, BMI z-scores decreased in the NG (*p* < 0.05); no between-group difference was found.
Leidy 2015 [16]	USA Setting not provided	Mean age: 19 ± 1 years Enrolled (*N* = 57) Breakfast-skipping (*n* = 9) Normal-protein breakfasts (*n* = 21) High-protein breakfasts (*n* = 24)	Assignment to one of three breakfast groups: (i) Normal-protein breakfasts (NP—13 g protein); (ii) High-protein breakfasts (HP—35 g protein); (iii) Continue to skip breakfast (CON).	Body weight Body composition 3-day free-living perceived appetite 3-day daily food intake	Compared to CON, HP showed the prevention of fat mass gains over the 12 weeks (*p* = 0.02), reductions in daily intake (*p* = 0.03), and reductions in daily hunger (*p* < 0.05).
Ochoa-Avilés 2017 [38]	Ecuador School	12–14 years old Enrolled (*n* = 1300) 10 intervention clusters (*n* = 702) 10 control clusters (*n* = 728)	Intervention-*ACTIVITAL* included: (i) Programmatic workshops with students and staff; (ii) Healthy-eating workshops with students and staff; (iii) Curriculum-based toolkit in classrooms; (iv) Environment-based staff and parent workshops; (v) Healthy breakfast preparation. Control: usual health and science lectures delivered in the school setting.	Dietary intake (two 24 h dietary recalls conducted on two randomly chosen weekdays): added sugar, fruit and vegetables, unhealthy snacking, consumption of unhealthy school snacks, and breakfast Waist circumference	Compared to control participants, intervention participants consumed lower quantities of unhealthy snacks (*p* = 0.04) and less added sugar (*p* = 0.006) at the end of the intervention. Compared to control participants, intervention participant waist circumference was lower at the end of the intervention (*p* = 0.005).
Shomaker 2019 [5] Bernstein 2021 [11] 1.5-year follow-up to Shomaker et al. (2019)	USA Clinical	12–17 years old Enrolled: (*n* = 54) Intervention (*n* = 29) Control: (*n* = 25)	Intervention: mindfulness-based education curriculums; six weekly one-hour sessions. Control: control-conditions health education program with same exposure.	BMI Percent body fat Perceived stress Food reward sensitivity Stress eating Executive function (BRIEF survey; NIH Toolbox Flanker Inhibitory Control and Attention Test; NIH Toolbox List Sorting Working Memory Test)	Intervention participants demonstrated lower food reward compared to control participants (*p* = 0.01) at 6 months. Intervention participants demonstrated reduction in stress eating compared to control participants (*p* = 0.05) at 6 months. No intervention effects for BMI or percent fat mass were observed.
					At 1.5 years, stress eating significantly increased in control participants compared to intervention participants (*p* = 0.01). No other intervention effects were observed.
Van Epps 2016 [15]	USA Online	12–18 years old Enrolled (*n* = 2202 No label (*n* = 378) Calorie label (*n* = 360) California warning label (*n* = 366) Weight-gain warning label (*n* = 366) Preventable warning label (*n* = 357) Type 2 diabetes warning label (*n* = 375)	Participants engaged in a hypothetical beverage-choice scenario and were randomized into one of six food-labeling groups: (i) No warning labels; (ii) Calorie label; (iii) California warning label; (iv) Weight gain warning label; (v) Preventable warning label; (vi) Type 2 diabetes warning label.	Beverage choice (hypothetical) Perceptions of beverages Interest in coupons Endorsement of warning label	In three of the four warning-label groups, participants chose sweetened beverages significantly less frequently than in the no-label group.
Multicomponent Interventions					
Anderson 2017 [33]	New Zealand Home	5–16 years old Enrolled (*n* = 203) Intervention (*n* = 69) Control (*n* = 69)	Both Intervention and Control groups received assessments, advice from a multidisciplinary team, and 6-month follow-up with home visits. Intervention group additionally participated in weekly group sessions delivered by a physical activity coordinator, dietitian, and psychologist.	BMI Health-related quality of life (HRQOL) Child Behavior Checklist (CBCL) Physical activity (steps/d, moderate to very vigorous physical activity) Cardiovascular fitness (550 m walk/run time) Screen time Hemoglobin A1c Fasting insulin	Both groups displayed improvements in BMI at 6 and 12 months, with no significant differences between groups. Intervention group demonstrated higher HRQOL scores (*p* = 0.013) and better CBCL scores (*p* = 0.032).
Barnes 2020 [34] Nutrition intervention component from Sutherland et al. [35]	Australia School	5–12 years old Enrolled (*n* = 815) Physical activity (*n* = 283) Nutrition (*n* = 163) Combined (*n* = 202) Control (*n* = 167)	4 treatment groups: (i) Physical activity—150 min of planned in-school physical activity; (ii) Nutrition—a healthy school lunch-box; (iii) Combined physical activity and nutrition; (iv) Control.	BMI Waist circumference Quality of life (QoL) (Paediatric Quality of Life Inventory (PedsQL))	Participants in the nutrition group had healthier BMIs (*p* = 0.02), while those in the physical activity group reported a lower waist circumference (*p* = 0.03). Neither the nutrition or physical activity intervention had a significant effect on child BMI scores or child quality of life. No significant synergistic effects were observed for the two interventions combined.
Sutherland 2019 [35]		5–12 years olds Enrolled (*n* = 1769) Intervention (*n* = 778) Control (*n* = 991)	4 treatment groups: (i) No intervention; (ii) Physical activity intervention only; (iii) Lunchbox intervention only; (iv) Physical activity and lunchbox intervention combined. The lunchbox intervention “SWAP IT” included: (i) School nutrition guidelines; (ii) Lunchbox lessons; (iii) Information pushed to parents via a school communication app; (iv) Parent resources addressing barriers to packing healthy lunchboxes.	Reduction in mean energy (KJ) packed in school lunchboxes	At 10 weeks, there were no significant differences between groups in the mean energy of foods packed within lunchboxes (*p* = 0.22). Participants who received intervention components demonstrated a significant increase in mean lunchbox energy from recommended foods (*p* = 0.04) and a non-significant increase in the percentage of lunchbox energy from recommended foods in intervention schools (*p* = 0.08).
Collins 2014 [32]	Australia School	Mean age: 13.2 ± 0.5 years Enrolled (*n* = 330) Intervention (*n* = 159) Control (*n* = 171)	*NEAT* girls’ physical activity and nutrition handbook: 10 weeks of health information and home challenges designed to promote healthy eating and physical activity.	Dietary intake was assessed using the Australian Child and Adolescent Eating Survey (ACAES) FFQ	There were no statistically significant grouped-by-time effects for dietary intake or food-related behaviors. 12-month trends suggested more intervention group girls had improved water intake (*p* = 0.052) and consumed < one sweetened beverage per day (*p* = 0.057).
Cullen 2013 [19]	USA Online	12–17 years Enrolled (*n* = 309) Intervention (*n* = 288) Control (*n* = 102)	Intervention - *Teen Choice: Food and Fitness* was an 8-week program delivered through an interactive website that consisted of: (i) Role model stories; (ii) Self-monitoring; (iii) Goal review; (iv) Problem-solving components.	Fruit and vegetable intake (Youth Risk Behavior Survey) Physical activity (Youth Risk Behavior Survey)	Significantly more intervention participants reported eating more vegetables compared to control participants (*p* < 0.05). Control participants reported significantly more physical activity enjoyment compared to intervention participants (*p* < 0.001).
DeBar 2016 [24]	USA Clinical	Female 12–17 year olds with BMI percentile ≥90th	Intervention: (i) Sixteen 90 min group meetings over 5 months; (ii) Tracking changes in dietary intake and eating patterns; (iii) Planning developmentally tailored physical activity; (iv) Discussing issues often associated with obesity in girls; (v) Training primary-care physicians to support goals; (vi) Weekly parent support meetings during first 3 months.	BMI Metabolic: total cholesterol, High-Density Lipoprotein (HDL), Low Density Lipoprotein (LDL, triglycerides, and fasting glucose Three 24 h dietary recalls 24 h physical activity recall Hours per week of screen time Average days per week breakfast consumed Average number of family meals per week Average number of times per week fast food and sugar-sweetened beverages consumed Health/lifestyle behaviors and utilization of weight-management services Psycho-social: (i) Eating and mood disorder symptoms (Questionnaire of Eating and Weight Patterns—Adolescent Version (QEWPA), Patient Health Questionnaire for Adolescents (PHQ-A)); (ii) Body satisfaction (Body Satisfaction Scale); (iii) Internalization of sociocultural attutdes toward appearance (Sociocultural Attitudes Towards Appearance Scale (SATAQ-3)); (iv) Self-esteem (Rosenberg Self-Esteem Scale); (v) Quality of life pediatric (Quality of Life Inventory (PedsQL)).	Intervention participants demonstrated significantly greater reductions in BMI z-scores over time (*p* = 0.01). No intervention effects for other metabolic factors were demonstrated. Intervention participants reported significantly greater body satisfaction (*p* = 0.03), less internalization of social norms regarding attractiveness (*p* = 0.02), less reduction in family meals (*p* = 0.03), and less fast food consumption (*p* = 0.02).
Duus 2022 [42]	Denmark School	15–20 years old Enrolled (*n* = 31) Intervention (*n* = 16) Control (*n* = 15)	Intervention - *Healthy High School* (HHS) included: (i) HHS behavior-change curriculum; (ii) HHS catalogue of organizational and environmental changes for supportive school environments; (iii) Peer-led innovation workshop and activities; (iv) Smartphone app.	Meal frequency Daily water intake Fruit intake Vegetable intake	No significant between-group differences were observed.
Freira 2018 [43]	Portugal School	14–19 years old Enrolled (*n* = 97) Intervention (*n* = 46) Control (*n* = 5)	Intervention and control groups both received: (i) Three 30 min lifestyle counseling sessions, 3 months apart; (ii) Individualized nutrition counseling; (iii) Physical activity exercise plan. Intervention (motivational interviewing group (MIG)) focus was collaboration, evocation, and autonomy. Control (conventional intervention group (CIG)) received conventional counseling style with provision of information, instruction, and advice.	BMI z-score Waist circumference Percent of fat mass Percent muscle mass Blood pressure	MIG participants demonstrated significant decreases in BMI z-score at both 3 and 6 months (*p* < 0.001); CIG participant BMI z-scores decreased non-significantly. MIG participants demonstrated a significant decrease in waist circumference, percent fat mass, and systolic and diastolic blood pressure and an increase in percent muscle mass at both 3 and 6 months (*ps* < 0.01); CIG participants demonstrated a significant increase in abdominal waist circumference, percent fat mass, and systolic and diastolic blood pressure and a decrease in percent muscle mass at both 3 and 6 months (*ps* < 0.001).
Lubans 2016 [31]	Australia School	12–14 years old Enrolled (*n* = 361) Intervention (*n* = 181) Control (*n* = 180)	Intervention - *ATLAS*: lasted 20 weeks and included: (i) Teacher professional learning; (ii) Provision of fitness equipment to schools; (iii) Researcher-led seminars; (iv) Face-to-face physical activity sessions; (v) Lunch-time physical peer-led activity leadership sessions; (vi) Pedometers; (vii) Parental newsletter for reducing screentime; (viii) Web-based smartphone application.	BMI Waist circumference Physical activity Sedentary behavior Sugar-sweetened beverage consumption Muscular fitness Resistance training skill competency Motivation for school sport	No intervention effects for BMI or waist circumference were observed. Intervention effects on screentime (*p* = 0.03), resistance-training skill competency (*p* < 0.01), and motivational regulations (intrinsic regulation (*p* = 0.03); identified regulation (*p* = 0.028); introjected regulation (*p* = 0.06)) for school sport were observed.
Mameli 2018 [40]	Italy Clinical	10–17 years old Enrolled (*n* = 43) Intervention (*n* = 16) Control (*n* = 14)	Intervention: (i) Low-energy diet; (ii) Physical activity prescription; (iii) Wristband and smartphone application with personalized feedback every 7 days. Control group: (i) Low-energy diet; (ii) Physical activity prescription.	Weight BMI Energy and macronutrient intake	No significant intervention effects were observed.
Pbert 2013 [13]	USA School	Grades 9–11 Enrolled (6 schools; *n* = 82 participants) Intervention (*n* = 42) Control (*n* = 40)	*Lookin’ Good Feelin’ Good* intervention: (i) School nurse intervention—6 weekly 30 min individual counseling sessions followed by a maintenance phase of 6 monthly sessions; (ii) After-school exercise program—3 sessions per week for 8 months. Control participants had 12 individual visits with the school nurse during non-academic lessons.	BMI Dietary intake (24 h dietary recall) Physical activity (ActiGraph Model GT1M): average daily minutes of light, moderate, and vigorous activity Sedentary behavior (2 items from the Youth Risk Behavior Survey) Other outcomes: sociodemographic variables, self-efficacy, and barriers to healthy eating and exercising	Compared to control participants, intervention participants ate breakfast on significantly more days/week, had a lower intake of total sugar, and had a lower intake of added sugar at 2 months. Compared to control participants, intervention participants were more likely to drink soda ≤ one time/day and eat at fast-food restaurants ≤ one time/week at 6 months. No significant differences between groups were observed for BMI, physical activity, or caloric intake.
Pbert 2016 [14] 8-month follow-up to Pbert et al. [13]		Grades 9–12 Enrolled (8 schools; *n* = 126) Intervention (*n* = 54) Control (*n* = 57)			No between-group differences were observed for BMI, percent body fat, and waist circumference at 8-month follow-up. Compared to control participants, intervention participants reported eating breakfast on more days/week (*p* = 0.024) and participating in more physically activity in the past 7 days (*p* = 0.007).
Robbins 2020 [9]	USA School and Home	10–13 years old Intervention (*n* = 39) Control (*n* = 45)	*Guys/Girls Opt for Activities for Life* (GOAL) 12-week intervention included: (i) After-school club 2 days/week; (ii) Parent–adolescent dyad meeting; (iii) Parent Facebook group. Control participants participated in usual activities.	Minutes of moderate-to-vigorous physical activity (MVPA; ActiGraph GT3X+) Diet quality (Automated Self-Administered 24-Hour Dietary Assessment Tool) Psychosocial perceptions related to PA and healthy eating	In the intervention group, autonomous motivation for physical activity and self-efficacy for healthy eating were significantly higher compared to the control group post-intervention (both *ps* < 0.05). No significant between-group differences were observed for fruit or vegetable intake, diet quality, MVPA, percent body fat, or BMI percentile.
Sgambato 2019 [36]	Brazil School and Home	Fifth and sixth graders Mean: 11.5 years Enrolled (*n* = 2743) Intervention group (*n* = 1164) Control group (*n* = 1112)	Intervention: (i) Six 50 min intervention sessions to encourage heathy eating and physical activity; intervention activities included educational games, group debates, and culinary classes; (ii) Community health agents visit families in their homes and integrate into a subgroup of participants.	BMI Percent body fat Dietary intake (FFQ) Physical activity (self-reported) participated in during the last 7 days (Brazilian National School-Based Health Survey (PeNSE))	Intervention participants demonstrated an increase in BMI compared to control participants (*p* = 0.05). Male intervention participants demonstrated a significantly greater decrease in % body fat (*p* = 0.03). Intervention participants showed significantly increased physical activity compared to control participants (*p* < 0.05). Female adolescents in the intervention group ate healthier items more frequently. Participants who received both in-school and at-home intervention components showed significantly increased percent body fat compared to the control participants (*p* = 0.01).
Vidmar 2019 [10]	USA Clinical	12–18 years Enrolled (*n* = 35) EMPOWER intervention (*n* = 17) App intervention (*n* = 18)	Intervention – *EMPOWER* included: (i) Monthly clinic visits; (ii) Individualized behavior-change goals for healthy eating, physical activity, emotional wellbeing, and family support. App intervention: (i) Two clinic visits at 3- and 6-month intervals; (ii) Text messages 5 days per week; (iii) Weekly phone sessions.	Mean change in zBMI and % BMI over 95th percentile (%BMI_p95)_	App participants demonstrated significant decreases in zBMI and %BMI_p95_ (*p* < 0.001 and *p* = 0.001) compared to EMPOWER participants (*p* = 0.31 and *p* = 0.06).
Viggiano 2013 [39]	Italy School	Middle- and high-school students *n* = 20 schools (10 intervention; 10 control) Enrolled (*n* = 3110) Intervention (*n* = 1663) Control (*n* = 1447)	Intervention—Kaledo nutrition and phyical activity-oriented board game [44]. 15–30 min game sessions.	BMI Adolescent Food Habit Checklist Dietary questionnaire	Significant intervention effects were observed according to the Adolescent Food Habit Checklist at 6 months (*p* < 0.001) but not at 18 months; school level moderated the relationship, in that greater differences were observed in high school compared to middle school (*p* = 0.02). Significant intervention effects for four sections of the dietary questionnaire were observed—nutrition knowledge (*p* < 0.001), healthy and unhealthy diet food (*p* < 0.001), food habits (*p* < 0.002), and physical activity (*p* < 0.001). Intervention effects manifesting as lower BMI z-scores were observed at 6 months (*p* = 0.001), and 18 months (*p* = 0.017).
Whittemore 2013 [20]	USA School and Online	14–17 years old Enrolled (*n* = 384) Intervention (*n* = 207) Control (*n* = 177)	Intervention - *HEALTH[e]TEEN*– included: (i) 8 lessons—nutrition, physical activity, metabolism, and portion control; (ii) Goal setting; (iii) Self-monitoring; (iv) Health coaching; (v) Social networking. *HEALTH[e]TEEN + CST*: aforementioned components plus an additional 4 lessons on coping-skills training.	BMI Sedentary behavior (self-reported questionnaire) Physical activity (Youth Risk Behaviors Survey) Nutrition behavior (22-item survey adapted from the After-School Student Questionnaire) Self-efficacy for healthy eating and physical activity (After-School Student Questionnaire)	Participants across groups demonstrated significant improvements in self-efficacy (*p* < 0.001), healthy eating behavior (*p* < 0.001), fruit and vegetable intake (*p* < 0.001), moderate and vigorous exercise (*p* < 0.001), stretching exercises (*p* < 0.01), sugar-sweetened drink intake (*p* < 0.001), junk-food intake (*p* < 0.01), and sedentary behavior (*p* < 0.001). A marginally significant decrease in weight (*p* = 0.05), but not BMI (*p* = 0.86), was observed; BMI z-scores and percentiles were not calculated. HEALTH[e]TEEN + CST participants experienced a significantly smaller increase in weight (*p* = 0.03) and BMI (*p* = 0.05).

## 4. Discussion

Overweight and obesity persist as medical and public health issues of concern across the globe [1]. Of particular concern are the high rates of overweight and obesity among children and adolescents, due to their detrimental impacts on physical, mental, and social health; academic success; and overall health and wellness. The adolescent developmental period is a particularly unique time for intervention, given the budding autonomy experienced at these ages. The objective of the current study was to systematically review the current peer-reviewed scientific literature to understand what approaches are effective at impacting adolescent obesogenic dietary behaviors, as tested through rigorous RCT designs.

More than half of the studies included in the current systematic review demonstrated significant improvements in the primary outcomes investigated. Dietary behavior improvements were observed in several studies, with the most frequently noted improvements being related to fruit consumption, sugar-sweetened beverage consumption, and snacking [5,13,14,18,19,20,22,23,26,27,32,34,35,38,39,41]. These findings are notable given the role of excessive energy intake, particularly sugar intake, and its relationship to excessive weight gain [45,46]. Twenty-five of the included studies assessed anthropometric and biometric outcomes, and eleven of these studies reported significant improvements among intervention participants, with the most frequently examined and reported measures being BMI and waist circumference [10,12,17,24,27,29,33,34,38,39,43]. Bagherniya et al. [29], DeBar et al. [24], Ochoaaviles et al. [38], Hidayanty et al. [27], and Viggiano et al. [39] reported significant improvements in both dietary outcomes and anthropometric outcomes. The remaining studies wherein significant anthropometric or biometric findings were reported did not report on dietary outcomes [10,12,17,33,34,43].

The setting of the interventions varied across the studies included in the current systematic review—school, home, clinical, or online. Of the studies wherein significant improvements were observed for dietary, anthropometric/biometric, or physical activity outcomes, twenty-one included a school intervention setting [12,13,14,17,18,20,25,26,27,28,29,31,32,34,35,37,38,39,41,42,43]. Moreover, all but one [24] of the studies that demonstrated significant improvements in more than one category of outcomes (i.e., dietary, anthropometric/biometric, or physical activity outcomes) were conducted in the school setting [13,14,20,27,29,34,35,38,39]. It is known that schools are an opportune setting for interventions with school-aged children, due to the captive-audience nature of the setting as well as the amount of time spent in the setting. Schools can provide access to healthy foods and snacks, spaces for safe and structured play, education, role-modeling, and peer support [47]. With regards to food and snack options, schools often provide meals and snacks as components of national child meal programs, which have nutritional standards (e.g., the United States Department of Agriculture National School Lunch Program). Schooldays also provide structure and routine, as posited by the Structured Days Hypothesis [48], which counter negative obesogenic behaviors. Schools may be a particularly advantageous setting for interventions targeted toward adolescents, given the growing impact of peer influence at this development stage. 

Leading professional, governmental, and international organizations have repeatedly recommended approaches to childhood overweight and obesity prevention that include multiple components, social and environmental levels, and systems approaches [49,50,51]. More than half of the high-impact studies were either multicomponent (addressing more than one behavior (e.g., diet and physical activity) or health outcome (e.g., stress and wellbeing)) or multilevel (intervening at more than one level of the socio-ecological model) in nature [13,14,17,18,20,22,23,24,27,29,31,34,35,38,39]. All of the studies that demonstrated significant improvements in more than one category of outcomes (i.e., dietary, anthropometric/biometric, or physical activity outcomes) were either multicomponent or multilevel in nature [20,24,27,29,38,39]. Despite the recommendations and apparent successes evidenced by this review, the implementation of multicomponent and multilevel interventions remains challenging with regard to resources, capacity, and sustainability. 

The present study has some limitations. First, as with any review of scientific literature, publication bias, the selective representation of the findings based on the research hypothesis, could have impacted the summary of the literature and the conclusions drawn across the studies included in the current systematic review. Specifically, the published literature may have been biased towards studies where only significant findings were demonstrated. The authors were limited to rigorously evaluating only studies that had been published in peer-reviewed research journals. Second, given the broad scope of the studies included, i.e., not restricted to a particular country, income group, setting, or type of intervention activity, there was great heterogeneity across the included studies, which may have limited our conclusions. Finally, process outcomes and exposure time were rarely and inconsistently reported, which posed a challenge for comparison across studies. 

## 5. Conclusions

Adolescence is a unique developmental period, in that the potential for health risks is high, but the emergence of autonomy and peer influence provides a unique opportunity to intervene to curb these health risks. Interventions targeting dietary behaviors in adolescents that are delivered in the school setting and are multicomponent or multilevel in nature are the most effective in terms of impact on dietary intake, anthropometric or biometric outcomes, and physical activity. Professionals in the field should continue to engage in the school setting despite the challenges that exist, e.g., additional approval processes and adapting intervention approaches to already-limited school hours and structures. The scientific community still needs to improve on the transparency of intervention approaches with regards to process outcomes, such as exposure time, the feasibility of intervention component delivery, and the acceptability of intervention components, in order to most effectively replicate and scale-up successful approaches to adolescent overweight and obesity prevention.

## Figures and Tables

**Figure 1 nutrients-14-04592-f001:**
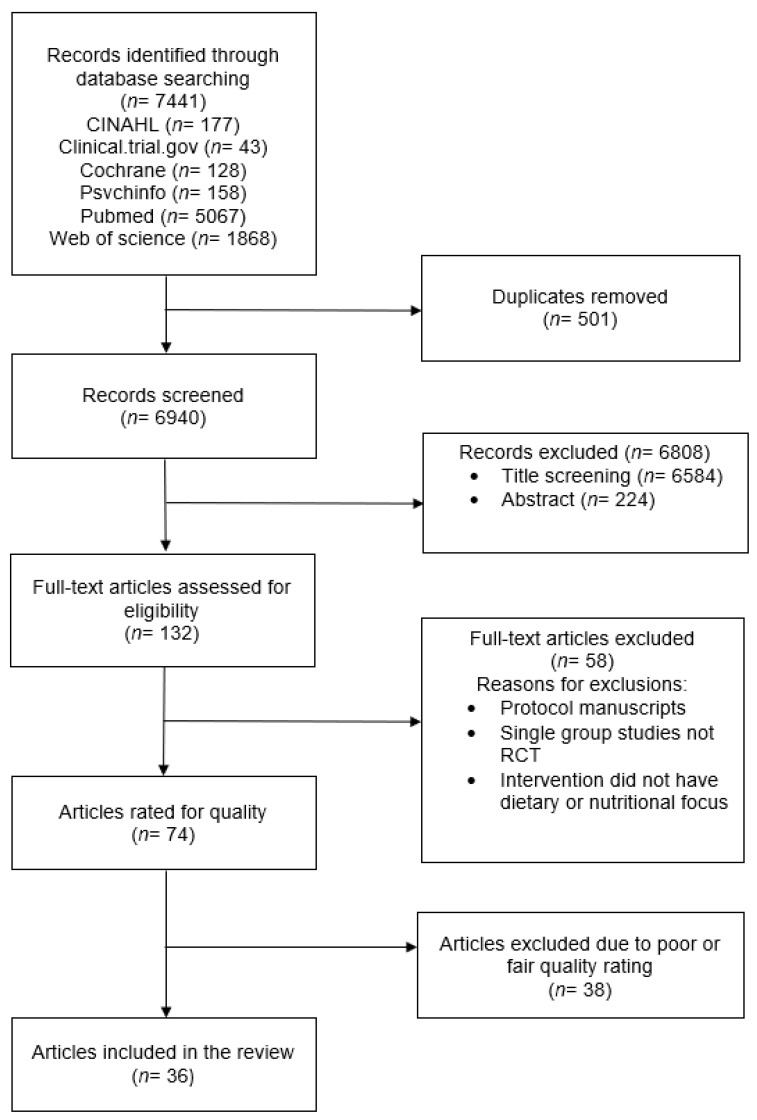
PRISMA diagram.

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
