# Peer review of "Improving Obesogenic Dietary Behaviors among Adolescents: A Systematic Review of Randomized Controlled Trials"

_nutrients, 2022, doi:10.3390/nu14214592_

Round 1

Reviewer 1 Report

Authors proposed a paper entitled “Improving Obesogenic Dietary Behaviors among Adolescents: A Systematic Review of Randomized Controlled Trials” for the publication in Nutrients, MDPI.

The paper has a quite good scientific soundness, but I suggest some revisions before publication.

I suggest adding an abbreviation list containing all the abbreviations and acronyms used in this paper.

Line 39. Is the [1] reference related also to line 37?

Lines 44 and 45, repetition of the world “conditions”. Please, avoid repetitions.

Line 52, Line 53 and Line 113 and Line 117, 121. Remove the double space here.

Line 67. Could it be better to say “up to 35 year?” or “until 35 years”?

Line 89. Could the author explain why they are referring to this fact? Are the not-in-English studies particularly numerous such to justify this comment?

Line 126. “=.” This is a spelling mistake. Please check it.

Line 127. “The Asian countries were the second most published”. I think this should be said differently: publications were proposed by Asian authors, or something like that.

Line 218. “Whittemore and colleagues” could be “Whittemore et al.”

Line 239. Double space.

Line 260. “Schools have the ability to provide access to healthy food and snacks, spaces for safe and structured play, education, role modeling…” could author develop this concept?

Line 268. “Of the studies that demonstrated significant impacts, approximately half of those”. I would say “Half of these high-impact studies, …”

Line 279. “publication bias” could the author better expand the concept of bias in publications?

Author Response

We thank you for your thorough review and comments.  Please see attached document with our responses to your specific comments. Thank you, again, for your time.

Reviewer 2 Report

1. Section 2.1: “web of science”? This is a key database.

2. Line 196-201 and Table 1: The labeling of references does not meet the requirements of journals

3. The first occurrence of the abbreviation should have a full name.

4. Fig 1: The paper screening method is too simple, and the screening basis needs to be refined. For a review, 41 references are obviously insufficient.

5. 3.3. Effect of interventions: The author needs to further optimize and review the results, especially the comparison between literatures, rather than stating some literatures.

6. Line 278: “Limitations” is not enough. A review should have a significant discussion on future research.

Author Response

(The authors gave the same response as above.)

Reviewer 3 Report

In the current manuscript, authors have presented a systematic review of randomized control trials that aimed to improve dietary intake and behaviors among adolescents. Based on the study, the most effective interventions are multicomponent or multilevel in nature and delivered in the school settings.

The article is well structured into section and subsections. English is clear and professional. It is within the scope of journal.

There are some minor comments to improve the article:

1)     Page 3, Line 117: There is a typing error. The “National Heart, Lung, and Blood Institute” is mentioned as, “Health National Heart, Lung, and Blood Institute”

2)     Page 3, Line 126: There is an unwanted symbol “=” in between sentences.

3)     Conclusion section, Line 294-297: The sentence needs to be reframed to improve clarity.

4)     References – The referencing format needs to be consistent and complete. Some have missing page numbers or unwanted symbols. Check reference number 1, 8, 10, 11, 25, 29, and 31.

Author Response

(The authors gave the same response as above.)

Round 2

Reviewer 1 Report

Authors proposed a revised version of their paper entitled “Improving Obesogenic Dietary Behaviors among Adolescents: A Systematic Review of Randomized Controlled Trials” for publication in Nutrients, MDPI.

Authors addressed all the issues raised by the reviewers.

An abbreviation list has been added correctly.

The use of English has improved significantly, avoiding repetitions and improving the quality of discussion.

Inclusion criteria were clearly explained.

For this reason, I declare that the manuscript can be published in the present form.

Reviewer 2 Report

Ok